# Long-Term Clinical Assessment of Two Modifications of Tunnel Technique in Treatment of Cairo Recession Type 1 in Lower Jaw

**DOI:** 10.3390/ijerph192416444

**Published:** 2022-12-08

**Authors:** Anna Skurska, Robert Milewski, Małgorzata Pietruska

**Affiliations:** 1Department of Periodontal and Oral Mucosa Diseases, Medical University of Białystok, ul. Waszyngtona 13, 15-269 Białystok, Poland; 2Department of Biostatistics and Medical Informatics Medical University of Białystok, ul. Szpitalna 37, 15-295 Białystok, Poland

**Keywords:** recession, modified coronally advanced tunnel, subepithelial connective tissue graft, collagen matrix

## Abstract

Aim: The aim of this study is to compare long-term results after using an MCAT (Modified Coronally Advanced Tunnel) with an SCTG (Subepithelial Connective Tissue Graft) or an MCAT with CM (Collagen Matrices) in the treatment of Cairo recession Type 1 in mandibular single-rooted teeth. Material and method: The study encompassed 80 recessions in 18 patients for whom an MCAT was combined with CM on one side of the mandible and with an SCTG on the contralateral one. The following clinical parameters were measured: gingival recession height (GR) and width (RW), probing depth (PD), clinical attachment level (CAL), width of keratinized tissue (KT), gingival thickness (GT), and mean (MRC). Results: The MRC on the CM- and SCTG-treated sides was 55.25% and 82.35%, respectively. The SCTG side had a significantly greater improvement in MRC, GR, RW, KT, and GT compared to the CM side. The five-year results were stable relative to one-year observations. Conclusions: Both methods of treatment enable the achievement of stable long-term clinical results. Application of subepithelial connective tissue grafts is more effective relative to clinical parameters.

## 1. Introduction

Procedures of gingival recession coverage are performed for two reasons. The most frequent cause of periodontist consultation is dentition aesthetics. The second reason is teeth hypersensitivity to thermal stimuli [1,2,3,4]. Exposure of tooth roots may as well lead to carious or non-carious lesions, or to both of the above-mentioned effects. Gingival recessions are a frequent finding within adult populations and their prevalence is approximately 80% of the population [4]. 

The ultimate aim of surgical recession treatment is to reestablish the aesthetic soft tissue contour with the gingival margin positioned coronally from the cemento-enamel junction (CEJ). In cases of thin phenotypes, gingival thickening and maintenance of its appropriate structure, color, and soft tissue alignment are equally important [4,5]. Improving gingival phenotype by its augmentation correlates with its stability and decreases the risk of recurrence of gingival recession [3].

The standard procedure of soft tissue augmentation in muco-gingival surgery is the utilization of a subepithelial connective tissue graft (SCTG) [6,7]. The use of autogenous material is connected with the necessity of having an additional treatment site, which becomes the reason for more intense pain reported by patients [5,6,8,9]. It is also of a high importance to indicate limitations resulting from the anatomy of a donor site, particularly in the need of conducting a few such procedures, and the risk of early and late complications related to bleeding from a donor site. 

Considering all of the above-mentioned, development of treatment techniques concentrates on introducing biomaterials that could replace autogenous grafts. One of the available biomaterials is collagen matrices (CM), which feature two main benefits–they shorten the procedure time and decrease patient’s discomfort by elimination of palatal injury [10,11]. 

Taking into consideration both physical and financial costs related to coverage of gingival recessions, it seems to be founded to introduce a procedure with long-term efficiency. This postulate is also underlined in the recommendations of the 10th European Workshop on Periodontology and a recently published Cochrane systemic review. The authors indicated the importance of conducting at least 5-year observations in this field [12,13]. Unfortunately, in the literature there is not sufficient data on systemic reviews including long-term evaluation of treatment techniques, however, recently there have been more and more randomized trials [14,15,16]. In an article published in 2019 [17], we presented twelve-month observations describing the effectiveness of an MCAT (Modified Coronally Advanced Tunnel) with an SCTG or CM in treatment of multiple mandibular gingival recessions. 

The aim of this study is to compare long-term results after using an MCAT with an SCTG or an MCAT with CM in the treatment of Cairo recession Type 1 in mandibular single-rooted teeth.

## 2. Material and Methods

Study design, population, inclusion criteria and surgical procedure were described in the article evaluating 1-year results [17]. Briefly only: 

The study was designed as a single-center, randomized, split-mouth, assessor-blind trial. A total of 20 patients (13 women aged from 20 to 56 and 7 men aged from 23 to 43) referred to the Department of Periodontal and Oral Mucosa Diseases, Medical University of Białystok were included in the study. Allocation of treatment sites to test and control sites was performed by means of a computer-generated randomization table. A total of 18 patients (12 women and 6 men) aged 20–56 reported to control examination 5 years after the procedure. 

The patients signed a consent form to participate in the study. The study was performed in accordance with the Helsinki Declaration of 1975, as revised in 2000, and was reviewed and approved by the local ethics committee (APK.002.174.2020). 

## 3. Clinical Examination

The same examiner (A.S.) used a periodontal probe (PCP UNC15, Hu-Friedy, Chicago, IL, USA) to conduct the clinical examinations at baseline, one year following, and five years following. The intra-examiner reproducibility for GR measurements was assessed. The inter-class correlation coefficient was 98%. To calibrate the examiner, five patients who were not participating in the trial and who had at least two contralateral teeth that had recessions were employed. Each patient’s four teeth were examined by the examiner twice, 48 h apart. If measurements taken at baseline and 48 h later were equal to millimeters at >90% level, then the calibration was approved.

For each recession defect, the following clinical parameters were measured: gingival recession height (GR; at mid-buccal aspect of the tooth from the CEJ to the most apical extension of the gingival margin), recession width (RW; at CEJ level), probing depth (PD; at mid-buccal aspect of the tooth from the gingival margin to the bottom of the sulcus), clinical attachment level (CAL; at mid-buccal aspect of the tooth from the CEJ to the bottom of the sulcus), keratinized tissue width (KT), and gingival thickness (GT). KT was measured from the most apical point of gingival margin to the muco-gingival junction (MGJ). GT measurement was performed with a K-file 25 ISO and a silicon marker positioned perpendicular to the gingival surface at the mid-buccal aspect of the tooth on a long axis, 3 mm apically from the gingival edge.

The nearest 0.5 mm was used to round off all measurements.

Additional computations were made in order to assess the treatment’s efficacy:-Recession reduction (GRred) is equal to GR 0 − GR 5y.-GR 0 − GR 5y/GR 0 × 100%, which is the formula for mean root coverage (MRC).-GT gain equals GT 5y − GT 0, and KT gain equals KT 5y − KT 0.

### 3.1. Surgical Procedure

All surgical interventions were carried out by one surgeon (M.P.) using the modified coronally advanced tunnel technique, as described by Zuhr, Fickl, Wachtel, Bolz, and Hürzeler [18], with a collagen matrix on one side or a subepithelial connective tissue graft on the opposite side of mandible. 

In the recipient site, a full-thickness flap up to the muco-gingival junction (MGJ) and a split-thickness flap above the MGJ were prepared. A subepithelial connective tissue graft harvested from the palate and collagen matrix (Mucoderm^®^, Botiss biomaterials, Berlin, Germany) was positioned at the CEJ or 1 mm below the CEJ and stabilized with resorbable monofilament 6-0 sutures (Biosyn^®^, Covidien, Ireland).

The SCTG and CM were covered with a coronally advanced flap and secured with sling sutures using 6-0 non-resorbable monofilament suture (Ethilon^®^, Ethicon, Bridgewater, NJ, USA). The sutures were removed 2 weeks post-op (Figure 1 and Figure 2).

### 3.2. Statistical Analysis

In statistical analysis, the Kolmogorov–Smirnov test together with the Lillefors amendment and the Shapiro–Wilk test were used to confirm that the distribution was normal. The distribution of quantitative variables was not determined to be normal. In order to compare ordinal or quantitative variables lacking normal distribution, the non-parametric U Mann–Whitney test was utilized. The dependent variables were contrasted using the Wilcoxon matched pairs test. At *p* < 0.05, the findings were deemed statistically significant. The computations were performed using the Statistica 12.0 software program (StatSoft, Tulsa, OK, USA). In addition, the difference between GR5 and GR1 as well as the association between post-op GT and post-op KT were examined using a Spearman’s rank order correlation coefficient. Additionally, a univariate linear regression analysis was carried out to calculate the difference between GR1 and GR5, as well as the association between post-op GT and post-op KT.

## 4. Results 

The mean root coverage on the CM side was 55.25% after 5 years, whereas on the SCTG side it was 82.35%. In relation to examinations conducted one year after the procedure, no statistical differences were found on both sides. 

Over the period of five years, statistically significant reductions of GR values were achieved on both sides in comparison with the initial examination. Five years after treatment, no differences were found relative to examinations conducted one year after the procedure. GR values were reduced from 1.95 ± 0.76 mm to 0.95 ± 1.08 mm on the CM side and from 1.94 ± 0.66 mm to 0.46 ± 0.95 mm on the SCTG side in 5-year observations. In addition, a significant decrease of RW values, from 2.97 ± 0.75 mm to 1.57 ± 1.56 mm on the CM side and from 3.04 ± 0.73 mm to 0.82 ± 1.53 mm on the SCTG side, was noted. There were significant differences in all parameters (GR, MRC, ARC, RW) between the CM and SCTG sides in 5-year examinations.

After a 5-year observation period, a significant increase of KT and GT were noted on both sides, and the results were stable relative to 1-year observations. Similar to 1-year outcomes, the results between the sides differed in a statistically significant manner. There were significant differences in KT and GT gains between the two sides in 5-year examinations. All clinical parameter changes have been presented in Table 1.

There was a statistically significant correlation between KT1 and GR1-GR5 (*p* = 0.014, R Spearman = 0.38), as well as between dKT1 and GR1-GR5 (*p* = 0.015, R Spearman = 0.38) in the SCTG group (Figure 3 and Figure 4).

Data analysis also showed that it is essential to have at least 1 mm KT width and a KT-width increase of at least 0.5 mm one year after the procedure in order to maintain stability of outcomes within five years following the SCTG procedure. 

The univariate linear regression analysis showed a significant impact of KT1 and dKT1 on stability of GR (GR5-GR1) (respectively: *p* = 0.022, Coef. = 0.1203 and *p* = 0.027, Coef. = 0.1225), and a significant impact of KT1 and dKT1 on GR5 (respectively: *p* < 0.001, Coef. = −0.4153 and *p* < 0.001, Coef. = −0.3773).

## 5. Discussion

The aim of our study was to present the long-term assessment of the results of multiple gingival recession treatment in mandibular single-rooted teeth using an MCAT combined with an SCTG or CM. To the best of our knowledge, this is one of the few pieces of research to the discuss the given subject area, including 5 years of observations. Both treatment methods resulted in the improvement of clinical parameters with regard to recession reduction and increasing of KT width and GT thickness. The results proved to be equally stable over time. There were no significant differences observed in the described parameters between the examinations conducted 1 year and 5 years following the treatment. However, comparison between 2 sides done 5 years post-op showed, similarly as 1 year post-op, statistically significant differences in the outcomes. Either recession reduction, keratinized tissue width, or gingival thickness was significantly greater after using a connective tissue graft. The importance of SCTG utilization was underlined in the systemic review from 2019 concerning the stability of recession coverage procedures in time. Based on at least two years of follow-up, the use of an SCTG was found to condition the maintenance of MRC, CRC, and KTW outcomes [19].

The results of our study cannot be directly compared with those of other authors due to significant differences in treatment protocol and duration of follow-up. Most of the available publications present the results of treatment of gingival recession in the maxilla or in the maxilla and mandible together. It appears that satisfactory treatment results are more difficult to obtain in mandibular recessions due to less favourable anatomical conditions [4,20]. Moslemi et al. [21] conducted a 5-year assessment of the effectiveness of gingival recession treatment using an SCTG and an acellular dermal matrix. The results were similar in terms of CRC and recession reduction in both groups, however, the KT width increased only after using an SCTG. McGuire and Scheyer [22] made a comparison between CM and CTG combined with a coronally advanced flap five years after treatment. Regardless of the treatment methods used, they found a reduction in recession dimensions, concluding that CM may be an acceptable alternative to CTG. The authors also observed stability of results over time, between examinations performed six months and five years after treatment. They also highlighted the fact that there was no reduction in the width of the keratinized gingiva below 3 mm in the study group. 

In the literature, the relationship between the initial quality of the keratinized gingiva and clinical outcomes after the application of various root coverage techniques has been widely discussed [23,24]. The publication by McGuire and Scheyer [22] lacks information on the initial width of KT. In our study, the initial width of KT was less than 1.5 mm in both groups. The KT width we obtained at the 1-year follow-up increased slightly, and four years later increased to 2.05 ± 0.90 mm on the CM side and 4.10 ± 1.61 mm on the SCTG side. Although the minimum adequate width of KT is not known, according to some authors, KT width >2 mm is responsible for good maintenance prognosis [25]. According to other researchers, an initial larger KT dimension “helps” to achieve better clinical outcomes after various root coverage procedures. Therefore, it seems that the initial width of the KT may be a predictor of the effectiveness of the technique, and a KT width of more than 2 mm may be a positive predictor of the stability of the gingival margin position over time [26,27]. Adequate width of the keratinized gingiva may also be an indication for a less traumatic surgical technique using CM.

While examining both the correlation and regression, we noticed the relation between postoperative increase in gingival thickness and reduction of recession size after applying CM. However, our 5-year observations indicate statistically significant relation between the width of keratinized gingiva and its increase one year after treatment, and the outcome stability in the 5-year observations after using an SCTG. Tavelli et al. [16] reported an interesting analysis assessing 12 years of results after applying an acellular dermal matrix (ADM) with a coronally advanced flap (CAF) or tunnel technique (TUN). It was stated that presence of gingival thickness ≥1.2 mm after 6 months was found to be a predictor for the stability of the gingival margin throughout the 12 years. Yet, conducting the analysis of these outcomes, we did not notice such a correlation after applying both CM and an SCTG. 

Based on short- and long-term observations of the tunnel technique with CTG in lingual recessions coverage, Mancini et al. [28] pointed to significant reduction of recession and increase in width and thickness of keratinized gingiva. Five years after the treatment, tissues were observed to be stable, and the slight shift of gingival margin was noticed to occur only in one case. Authors also observed that modification of gingival phenotype after the procedure with CTG_enabled patients to maintain proper oral hygiene. 

With regard to the literature data, it should be stated that currently available collagen matrices cannot be considered as an equal valuable substitute for autologous tissue in terms of effectiveness of coverage procedures [11]. However, they are also said to bring some benefits–they shorten surgery and recovery time and decrease patient’s pain burden [10,11]. The use of CM should also be considered especially in cases where there are contraindications for the procedure of autologous CTG harvesting from the palate, or when the patient and the clinician wish to reduce biological costs and postoperative complaints and are willing to accept the probability of achieving a less than optimal result [11].

Analyzing the presented results, it is important to make a reference to their limitations. The main limitation is the conduction of a follow-up examination five years after surgery in an incomplete group, as two patients did not attend the visit. The weak point of our research is also a lack of information, which in accordance to some authors, may influence treatment results. We had no information on methods of home oral hygiene or supportive periodontal treatment. Regular periodontal supportive treatment, in particular, close surveillance of oral hygiene habits, is considered to be the main determinant of long-term success [19,29,30]. Moslemi et al. [21] showed that inappropriate horizontal brushing technique may increase the risk of gingival recession recurrence up to 11 times. However, the presented results make us believe that patients’ knowledge during the initial phase of treatment before the recession coverage procedures was partly effective and contributed to obtaining stable outcomes. 

## 6. Conclusions

Taking into account the limitations of our research, it can be concluded that both methods of treatment, MCAT + SCTG and MCAT + CM, enable the achievement of stable long-term clinical results. Using a subepithelial connective tissue graft is, however, more effective relative to reducing the size of gingival recession and improving the quality of soft tissues in terms of increasing the KT width and soft tissue thickness. 

## Figures and Tables

**Figure 1 ijerph-19-16444-f001:**
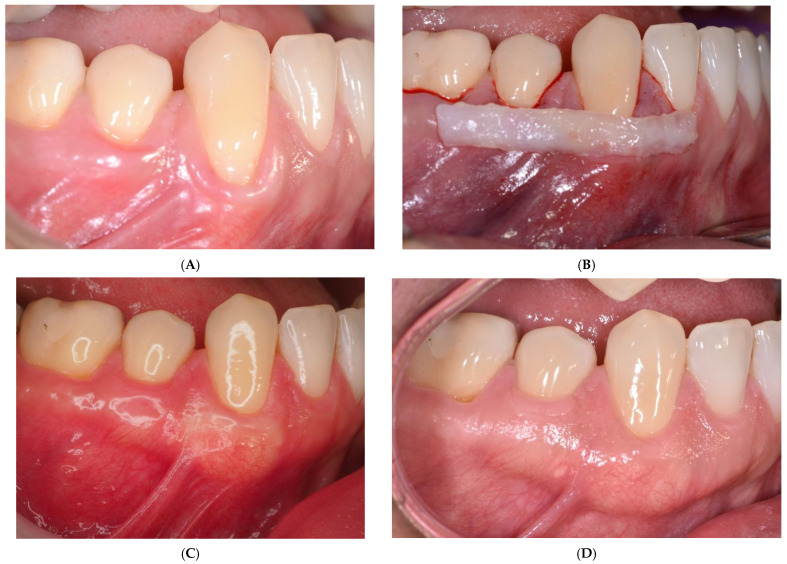
(**A**) Baseline SCTG side: the canine and the second premolar with recessions on the right side in the lower jaw; (**B**) position of the SCTG; (**C**) postoperative (12 months) view; (**D**) postoperative (5 years) view; (**E**) baseline CM side: the canine and the second premolar with recessions on the left side in the lower jaw; (**F**) CM position; (**G**) postoperative (12 months) view; (**H**) postoperative (5 years) view.

**Figure 2 ijerph-19-16444-f002:**
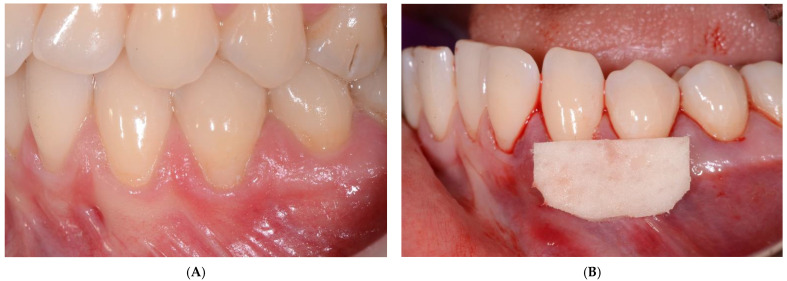
(**A**) Baseline CM side: the canine and the first premolar with recessions on the left side in the lower jaw; (**B**) CM position; (**C**) postoperative (12 months) view; (**D**) postoperative (5 years) view; (**E**) baseline SCTG side: the canine and the first premolar with recessions on the right side in the lower jaw; (**F**) SCTG position; (**G**) postoperative (12 months) view; (**H**) postoperative (5 years) view.

**Figure 3 ijerph-19-16444-f003:**
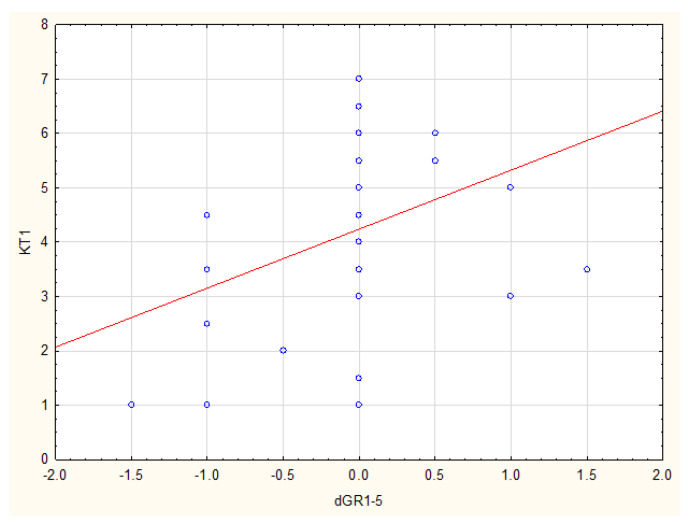
Correlation between KT width one year after procedure and GR changes between examination after one year and five years on the SCTG side.

**Figure 4 ijerph-19-16444-f004:**
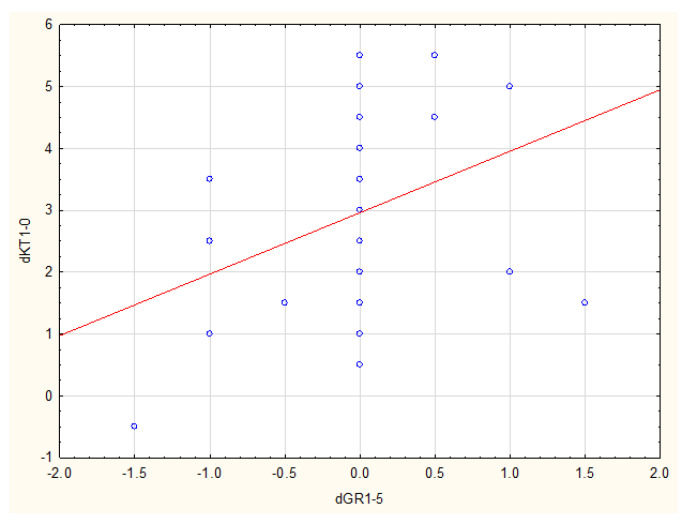
Correlation between KT-width increase one year after procedure and GR changes between examination after one year and five years on the SCTG side.

**Table 1 ijerph-19-16444-t001:** Clinical parameters (mean and SD) at baseline and at 1 year and 5 years post-surgery.

	Baseline	1 Year	P (0 v 1)	5 Years	P (0 v 5)	P (1 v 5)
GR SCTG	1.94(0.66)	0.40(0.69)	<0.001	0.46(0.95)	<0.001	0.556
GR CM	1.95(0.76)	0.95(0.79)	<0.001	0.95(1.08)	<0.001	0.919
P	0.7004	<0.001		0.0074		
MRC SCTG		83.10(27.63)		82.35(33.59)		0.861
MRC CM		53.20(32.17)		55.25(41.96)		0.742
P		<0.001		0.0026		
ARC SCTG		1.54(0.58)		1.47(0.69)		0.556
ARC CM		1.00(0.69)		0.93(0.84)		0.919
P		<0.001		<0.001		
RW SCTG	3.04(0.73)	0.89(1.37)	<0.001	0.82(1.53)	<0.001	0.924
RW CM	2.97(0.75)	2.08(1.30)	<0.001	1.57(1.56)	<0.001	0.030
P	0.4675	<0.001		0.0270		
PD SCTG	1.57(0.48)	1.58(0.64)	0.9546	1.70(0.46)	0.243	0.022
PD CM	1.47(0.46)	1.37(0.58)	0.3809	1.32(0.52)	0.291	1.0000
P	0.3135	0.0703		<0.001		
CAL SCTG	3.52(0.75)	1.98(0.88)	<0.001	2.17(1.01)	<0.001	0.037
CAL CM	3.43(0.93)	2.33(0.89)	<0.001	2.27(1.19)	<0.001	0.965
P	0.3055	0.0545		0.8195		
KT SCTG	1.28(0.72)	4.06(1.59)	<0.001	4.10(1.61)	<0.001	0.244
KT CM	1.38(0.68)	1.91(0.84)	<0.001	2.05(0.90)	<0.001	0.189
P	0.5909	<0.001		<0.001		
KT gain SCTG		2.78(1.53)		2.82(1.51)		0.244
KT gain CM		0.52(0.65)		0.65(0.76)		0.189
P		<0.001		0.002		
GT SCTG	0.76(0.31)	1.86(0.48)	<0.001	1.75(0.55)	<0.001	0.252
GT CM	0.82(0.30)	1.10(0.37)	<0.001	1.00(0.27)	0.006	0.247
P	0.2956	<0.001		<0.001		
GT gain SCTG		1.10(0.54)		1.02(0.64)		0.252
GT gain CM		0.27(0.40)		0.18(0.33)		0.247
P		<0.001		<0.001		
PI SCTG	0.02(0.07)	0.05(0.11)	0.2393	0.09(0.18)	0.036	0.289
PI CM	0.03(0.10)	0.04(0.09)	0.8139	0.08(0.14)	0.054	0.088
P	0.6992	0.7961		0.9352		
MBI SCTG	0.005(0.03)	0.04(0.10)	0.0499	0.02(0.07)	0.224	0.554
MBI CM	0.01(0.06)	0.005(0.03)	0.3613	0.01(0.06)	0.361	0.361
P	0.3037	0.0295		0.7014		
BOP SCTG	0.04(0.13)	0.07(0.13)	0.3202	0.04(0.09)	0.8067	0.202
BOP CM	0.06(0.14)	0.03(0.08)	0.3636	0.03(0.09)	0.444	0.767
P	0.5641	0.1145		0.7690		

## Data Availability

All data are available in the article.

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
