# Peer review of "Long-Term Clinical Assessment of Two Modifications of Tunnel Technique in Treatment of Cairo Recession Type 1 in Lower Jaw"

_ijerph, 2022, doi:10.3390/ijerph192416444_

Round 1
Reviewer 1 Report
The manuscript entitled "Long-term clinical assessment of two modifications of tunnel technique in the treatment of Cairo recession Type 1 in the lower jaw." describes the evolution after 5 years of 20 patients having different surgical approaches in treating type 1 recessions in lower dental arches.
While the study is pertinent, some modifications must be performed in order to be taken into consideration for publication:
1- the abstract style, and design style of the manuscript, including the divisions of the manuscript should be modified according to the journal's template; In the abstract, all the acronyms presented for the first time to the reader should be preceded by their full explanations, otherwise, any reader would be confused;
2 - Methods section should be modified drastically. While I understand this manuscript is a 5-year follow-up of the patients included in a study already published, all the information regarding the design of this study, the two surgical techniques, and the measurements performed afterward should be exposed here.
3-The conclusions should express in more detail the findings of this study, also related to the techniques used in the previous research the authors performed.
Author Response
Response to Reviewer 1 Comments
Reviewer: English language and style are fine/minor spell check required
Our respond: Thank you for this comment. The manuscript was edited by native speaker.
Reviewer: The abstract style, and design style of the manuscript, including the divisions of the manuscript should be modified according to the journal's template; In the abstract, all the acronyms presented for the first time to the reader should be preceded by their full explanations, otherwise, any reader would be confused;
Our respond: Thank you for this comment. The manuscript was edited.
Reviewer: Methods section should be modified drastically. While I understand this manuscript is a 5-year follow-up of the patients included in a study already published, all the information regarding the design of this study, the two surgical techniques, and the measurements performed afterward should be exposed here.
Our respond: Thank you for this comment. The manuscript in Method section was edited according to suggestions.
Reviewer: The conclusions should express in more detail the findings of this study, also related to the techniques used in the previous research the authors performed.
Our respond: Thank you for this comment. The manuscript was edited according to suggestions.
Reviewer 2 Report
This manuscript evaluate the long term results of gingival recession in mandibular single root teeth after the treatment of MCTA combined with SCTG or CM for five years follow-up. The experiments are well-designed and the results support the conclusion.
The topic may be of interest in the filed of Dentistry. Good paper, but seems like it should be submitted to a dental journal.
Miner comments;
1. Abstract. Please spell out MCAT, SCTG, MCAT, and CM when it first appears.
2. Conclusion. Please describe what is "both methods".
Author Response
Response to Reviewer 2 Comments
Reviewer: Abstract. Please spell out MCAT, SCTG, MCAT, and CM when it first appears.
Our respond: Thank you for this comment. The manuscript was edited.
Reviewer: Conclusion. Please describe what is "both methods".
Our respond: Thank you for this comment. The manuscript was edited according to suggestions.
Round 2
Reviewer 1 Report
The manuscript was improved by the modifications performed. I recommend accepting for publishing